# Player-optimal Stable Regret for Bandit Learning in Many-to-one Matching Markets with Substitutability

## Abstract

Bandit learning in matching markets has gained increasing attention, where one side of participants (players) learns unknown preferences through repeated interactions with the other side (arms). While prior studies mainly address one-to-one settings, many real-world applications — such as online advertising and negotiation between suppliers and demanders — naturally involve many-to-one matchings. Under the widely adopted substitutability condition, which guarantees the existence of stable matchings, learning becomes more challenging: players struggle to discover opportunities to be accepted by desirable arms due to the complex, set-dependent nature of arm preferences. Existing studies in this setting provide regret guarantees only for the player-pessimal stable matching, where the player side receives the least favorable outcome among all stable matchings. In this work, we propose a new algorithm: RIFLE, that addresses these limitations via a randomized initialization to uncover indexable preferences and an index-based phase of identifying explorable arms with decentralized conflict-free exploring, tailored for substitutable many-to-one environments. We theoretically prove that RIFLE converges to the player-optimal stable matching with a cumulative regret bound of $O(\max\{N, K\} \log T / \Delta^2)$, where $N$ is the number of players and $K$ is the number of arms. This result makes two key contributions. First, our approach is more general: it operates under the most general preference — substitutable preference conditions without pre-setting arm index. Second, we derive a player-optimal stable regret bound that is currently the best-known for both one-to-one and many-to-one matching markets. Empirical evaluations demonstrate that our approach significantly outperforms existing baselines in both matching quality and convergence speed.

## 1 Introduction

Matching markets have a rich research history in economics and social sciences, with a wide range of applications such as labor markets, college admissions, and online platforms (Roth, 1984; Gale & Shapley, 1962; Abdulkadiroğlu & Sönmez, 1999; Epple et al., 2006; Fu, 2014). In such markets, participants on each side typically maintain preference rankings over the other side. A central concern is market stability, since any unstable matching may unravel the entire system (Roth & Sotomayor, 1992). In many real-world scenarios—such as allocating advertising slots to advertisers or selecting an exclusive supplier for the product component—one side (e.g., platforms, product company) often handles multiple assignments simultaneously, leading to many-to-one matching settings. A rich body of work addresses how to compute stable matchings in these markets when participants' preferences are known in advance (Gale & Shapley, 1962; Roth & Sotomayor, 1992). However, in dynamic, short-term platforms (such as online advertising transactions or demander-supplier negotiations), agents often enter the market with uncertain preferences due to limited information about counterparty attributes or task relevance. Through repeated interactions, these preferences gradually become more refined, enabling participants to learn and adapt their matching choices over time.

The multi-armed bandit (MAB) framework is a classical model that captures the learning dynamics of a decision-maker in an uncertain environment (Lattimore & Szepesvári, 2020). Its application to bilateral matching markets was first proposed by Das & Kamenica (2005) and later formalized

theoretically by Liu et al. (2020). In this context, the problem is modeled as a platform where $N$ players aim to match with $K$ arms with unknown preferences. Through repeated interactions, players learn these preferences in order to identify a stable matching, which is framed as minimizing cumulative stable regret.

Extensive research has aimed to improve stable regret guarantees and generalize model assumptions. Liu et al. (2020) first study a centralized setting in which a central platform assigns arms to players, and they establish a player-pessimal stable regret guarantee—i.e., the outcome corresponds to the stable matching where each player is matched with their least preferred stable partner. Subsequent work (Liu et al., 2021; Kong et al., 2022) extends this to decentralized markets while also maintaining the player-pessimal stability guarantee. Other studies (Sankararaman et al., 2021; Basu et al., 2021; Maheshwari et al., 2022) assume a specific preference structure that ensures the existence of a unique stable matching. More recent advances (Zhang et al., 2022; Kong & Li, 2023; Kong et al., 2024) propose algorithms that achieve the player-optimal stable matching, where each player is matched with their most preferred stable partner.

Despite the significance of prior work, all existing studies assume that each market participant can be matched with at most one partner. However, in real-world applications, such as online labor markets, advertising matching platforms, or product supply companies usually have multiple slots or capacities, allowing them to match with multiple participants simultaneously. Wang et al. (2022) and Zhang & Fang (2024) attempt to extend existing algorithms to the many-to-one setting. They assume that the preference structure of arms remains the same as in the one-to-one setting, with the only difference being that each arm can accept its top-$C$ candidates, where $C$ denotes the arm's capacity (often referred to as the responsive preferences). In practice, however, the arms' preferences may be more complex and defined over combinatorial structures. For example, a platform may prefer a group of candidates with diverse characteristics, or a firm may value a balanced candidate cohort. Substitutability is among the most general combinatorial preference conditions under which stable matchings are guaranteed to exist. Yet, under such general preferences, only Kong & Li (2024) propose an algorithm that ensures a player-pessimal stable regret guarantee.

Efficient learning under substitutable preferences is fundamentally more challenging than in settings with responsive preferences. In the latter, a player can typically identify a stable preference ranking to guide exploration. However, with substitutable preferences, an arm's acceptance decision is not determined by a fixed order but is dynamically contingent on the entire set of competing players. This combinatorial complexity means that only a fluctuating subset of arms may be willing to accept a particular player at any given time. As a result, many exploration strategies become ineffective, as it is infeasible to construct exploration combinations that can guarantee acceptance across all arms. The learning challenge is thus elevated from identifying preferred arms to discovering viable application combinations in a dynamic environment. Furthermore, the aforementioned methods universally rely on a pre-assigned index for each arm, an assumption that often does not hold in practical applications. To address these challenges, we propose RIFLE (Randomized Index-Free Learning of Explorable Arms), a method combining randomized initialization with an index-guided phase that identifies explorable arms and coordinates decentralized conflict-free exploration. This mechanism ensures that all players with potentially stable partners systematically participate in exploration while avoiding conflicts with players without any stable partners. We prove that RIFLE converges to the player-optimal stable matching and achieves a stable regret upper bound of $O(\max\{N', K'\} \log T / \Delta^2)$, where $T$ is the horizon, $\Delta$ is the players' minimum preference gap, $N' \leq N$ denotes the maximum number of players that can be accepted when selecting an arm alone, and $K' \leq K$ denotes the maximum number of arms that a player can be matched when proposing alone. This result significantly improves existing results in both matching quality and regret rate. By providing a more general framework that accommodates the most general preference conditions without the assumption of a known arm index, our approach successfully establishes a new state-of-the-art performance benchmark in both one-to-one and many-to-one markets. Table 1 summarizes the regret bounds achieved by our method and compares them with those of existing algorithms.

## 2 RELATED WORK

The matching market model has been extensively studied in the literature (Roth, 1984; Roth & Sotomayor, 1992). One of the most fundamental settings is the one-to-one matching market, where

each participant can be matched with at most one partner from the other side. In this setting, each participant has a preference ranking over individual candidates, and the Gale–Shapley algorithm provides an efficient method for computing a stable matching that is optimal for the proposing side (Gale & Shapley, 1962).

Table 1: Comparisons of settings and regret bounds with most related works. # represents the centralized setting. $N, K, \Delta, c_j, C, \epsilon$ are the number of players and arms, some preference gap among players and arms, arm $a_j$'s capacity the total capacities of all arms under the responsiveness condition, and the hyper-parameter of algorithms which can be very small, respectively. We define $\Delta$ as the minimum preference gap relevant to the analysis, and specify $gap_1 - gap_5$ respectively as: the minimum preference gap between any two distinct arms across all players, the minimum gap among the top $min\{N + 1, K\}$ ranked arms of each player, the minimum gap between the player-optimal stable arm and its next preferred arm, the minimum gap among arms ranked above the one following the player-optimal stable arm, and the minimum gap between the player-pessimal stable arm and its next preferred arm. Based on the fact that the player-optimal stable arm must be the first $min\{N, K\}$-ranked, it holds that $gap_3 > gap_4 > gap_2 > gap_1$ and $gap_5 > gap_1$.

| Reference | Regret Bound | Setting and Regret Type |
|---|---|---|
| Zhang et al. (2022) | $O\left(\frac{K}{\Delta^2} \log T\right), \ N \leq K$ | One-to-one, Optimal, $gap_1$ |
| Kong & Li (2023) | $O\left(\frac{K}{\Delta^2} \log T\right), \ N \leq K$ | One-to-one, Optimal, $gap_2$ |
| Kong et al. (2024) | $O\left(\frac{N^2}{\Delta^2} \log T + \frac{K}{\Delta} \log T\right), \ N \leq K$ | One-to-one, Optimal, $gap_4$ |
| Sankararaman et al. (2021) | $O\left(\frac{NK}{\Delta^2} \log T\right)$ $\Omega\left(\frac{N}{\Delta^2} \log T\right)$ | one-one (serial dictatorship), $gap_3$ |
| Wang et al. (2022) | $O\left(\frac{NK^3}{\Delta^2} \log T\right) \#, \ N \leq C$ | Responsive, Pessimal, $gap_1$ |
| | $O\left(\frac{N^5 K^2}{\epsilon^{N^4} \Delta^2} \log^2 T\right), \ N \leq C$ | Responsive, Pessimal, $gap_1$ |
| | $O\left(\frac{K}{\Delta^2} \log T\right) \#, \ N \leq C$ | Responsive, Optimal, Known $\Delta$, $gap_3$ |
| Zhang & Fang (2024) | $O\left(\frac{\max(K,N)}{\Delta^2} \log T\right) \#$ | Responsive, Optimal, $gap_1$ |
| Kong & Li (2024) | $O\left(\frac{K}{\Delta^2} \log T\right) \#, \ N \leq K \cdot \min_{j \in [k]} c_j$ | Responsive, Optimal, $gap_2$ |
| | $O\left(\frac{N \min\{N,K\} C}{\Delta^2} \log T\right), \ N \leq C$ | Responsive, Optimal, $gap_4$ |
| | $O\left(\frac{NK}{\Delta^2} \log T\right)$ | Substitutable, Pessimal, $gap_5$ |
| **Ours** | $O\left(\frac{\max(K',N')}{\Delta^2} \log T\right)$ | Substitutable, Optimal, $gap_1$ |

Recognizing that participants in real-world markets often match with multiple partners, subsequent work has extended the model to many-to-one and many-to-many matching markets. However, stable matchings may not exist when participants have general combinatorial preference structures. In the many-to-one setting, the concepts of responsiveness and substitutability have been widely studied as sufficient conditions to guarantee the existence of stable matchings. Under these conditions, the Gale–Shapley algorithm can still be applied to compute a stable matching that is optimal for the proposing side (Roth & Sotomayor, 1992).

The bandit learning framework is first introduced to one-to-one matching markets to learn agents' unknown preferences and to converge toward stable matchings (Das & Kamenica, 2005; Liu et al., 2020). Existing works have made significant progress in improving the stability regret bounds, ranging from pessimal stable regret (Liu et al., 2020; 2021; Kong et al., 2022) and unique stable regret (Sankararaman et al., 2021; Basu et al., 2021) to optimal stable regret (Zhang et al., 2022; Kong & Li, 2023; Kong et al., 2024), thereby advancing both the theoretical understanding and practical performance of learning-to-match algorithms. The state-of-the-art results for player-optimal stable

regret are $O(K \log T/\Delta^2)$ in Zhang et al. (2022); Kong & Li (2023) and $O(N^2 \log T/\Delta^2)$ in Kong et al. (2024), each being optimal under different market scales.

Wang et al. (2022) extend existing algorithms from one-to-one to many-to-one matching markets with responsive preferences, and primarily establish player-pessimal stable regret guarantees. Although Zhang & Fang (2024) further improves these results by achieving an $O(\max\{N, K\} \log T/\Delta^2)$ regret value for optimal stable player matching, they directly let the player explore in order, which is hardly decentralized. Kong & Li (2024) consider the more general setting of substitutable preferences and derive a regret bound of $O(NK \log T/\Delta^2)$ for player-pessimal stable matchings. However, the above methods generally assume that each arm is associated with a predefined index, which is not a valid assumption in real-world settings. To the best of our knowledge, obtaining player-optimal stable regret guarantees in this more general setting without pre-setting the arm index remains an open problem.

Apart from this line of research, bandit learning in matching markets has also been explored in several other directions. These include settings where participants' preferences may involve indifferences (Kong et al., 2025; Lin et al., 2024), the incorporation of contextual information (Li et al., 2022), non-stationary preferences (Muthirayan et al., 2023; Ghosh et al., 2022), and scenarios with two-sided uncertainty (Pagare & Ghosh, 2023), where the preferences of both sides are initially unknown. Furthermore, recent work has considered notions of two-sided fairness (Hosseini & Zhang, 2024), as well as alternative learning objectives beyond regret, such as sample complexity guarantees for achieving stable stable outcomes (Hosseini et al., 2024).

## 3 PROBLEM SETTING

The market contains two sides of agents. One side can be modeled as players $\mathcal{N} = \{p_1, \ldots, p_N\}$ and the other side can be modeled as arms $\mathcal{K} = \{a_1, \ldots, a_K\}$. We consider the many-to-one matching markets, where each player (e.g., job seekers, demanders) can select one arm at each time step, while each arm can simultaneously accept multiple players. To systematically observe and analyze this problem, we introduce discrete time steps $t \in [T]$ as the basic time unit, where $T$ represents the total horizon.

Each agent in the market possesses a preference ranking over agents on the opposite side. Specifically, the preference of player $p_i$ for arm $a_j$ is quantified by a preference value $\mu_{i,j} > 0$, which reflects the player's overall satisfaction. A higher value indicates a stronger preference; that is, $\mu_{i,j} > \mu_{i,j'}$ implies that player $p_i$ prefers $a_j$ over $a_{j'}$. Following previous works, we assume the preference values towards different arms are distinct. Similar to real applications such as labor markets or advertising platforms where workers/advertisers often face uncertainty about their true preferences due to incomplete information including management style or conversion rates, players are uncertain about their preferences values and would learn them through repeated matching processes. On the arm side, although each arm $a_j$ can simultaneously accommodate multiple players, it does not accept all applicants indiscriminately due to capacity constraints. We introduce the notation $\text{Ch}_j(\cdot)$ to represent the preference behavior of arm $a_j$. When a set $S$ of players applies to arm $a_j$, it accepts its most preferred subset, denoted by $\text{Ch}_j(S)$. Similar to previous works on matching markets (Kong & Li, 2023; Liu et al., 2021; Wang et al., 2022; Kong & Li, 2024), we consider the setting where each arm knows its choice function which is determined by the observable characteristics of the players.

In each step $t \in [T]$ of the matching process, each player $p_i$ first submits an application to an arm $A_i(t) \in \mathcal{K}$. For any arm $a_j$, the applicant set $A_j^{-1}(t) = \{p_i : A_i(t) = a_j\}$ denotes all players proposing to arm $a_j$. The arm would accept its most preferred subset $\text{Ch}_j(A_j^{-1}(t))$. If the player $p_i$ is successfully matched with the selected arm, then it receives a reward $X_i(t)$ characterizing its satisfaction over this matching experience. Without loss of generality, $X_i(t)$ can be regarded as a 1-subgaussian random variable with expectation $\mu_{i,A_i(t)}$. And if $p_i$ is rejected by the proposed arm, it only receives $X_i(t) = 0$. For convenience, define $\bar{A}(t) = \{\bar{A}_i(t), i \in [N]\}$ as the matching outcome of round $t$, i.e., $\bar{A}_i(t) = A_i(t)$ if $p_i$ is successfully accepted and $\bar{A}_i(t) = -1$ otherwise.

Consider that there exists a player-arm pair $(p_i, a_j)$ such that:

- Player $p_i$ prefers $a_j$ over its current partner in $\bar{A}(t)$, i.e., $\mu_{i,j} > \mu_{i,\bar{A}_i(t)}$.

- The arm $a_j$ would accept $p_i$ when this player applies to $a_j$ together with its current partners, i.e., $p_i \in \text{Ch}_j \left( \bar{A}_j^{-1}(t) \cup \{p_i\} \right)$.

In this case, $p_i$ and $a_j$ have the incentive to deviate from their current partners and form a new matching pair. Such pairs are called blocking pairs. The matching $\bar{A}(t)$ is stable if no such blocking pair exists. It is worth noting that the stable matching may not always exist in the market. In this paper, we consider the scenario where arms' preferences satisfy substitutability (Roth & Sotomayor, 1992), which is one of the most general conditions to ensure the existence of a stable matching in the many-to-one markets.

**Definition 3.1** (Substitutable Preferences). For each arm $a_j$, its choice function $\text{Ch}_j(\cdot)$ satisfies substitutability if $\text{Ch}_j(S) \setminus \{p_i\} \subseteq \text{Ch}_j(S \setminus \{p_i\})$ for all $S \subseteq \mathcal{N}$ and $p_i \in S$.

Intuitively, this property states that arms view players as substitutes rather than complements: removing any player $p_i$ does not invalidate the acceptance of others in $\text{Ch}_j(S)$. For example, if $\text{Ch}_j(\{1, 2, 3\}) = \{1, 2\}$, then $\{1, 2\} \subseteq \text{Ch}_j(\{1, 2\})$ must hold even after removing player 3.

There may be more than one stable matching when arms have substitutable preferences. Define $\mathcal{M}$ as the set of all stable matchings. We focus on the player-optimal stable matching $\bar{m} \in \mathcal{M}$, which guarantees each player to be matched with their most preferred arm among all stable matchings, i.e., $\bar{m}_i \in \arg\max_{m \in \mathcal{M}} \mu_{i,m_i}$. The objective is to minimize the player-optimal stable regret for each player $p_i$ over $T$ rounds which is given by

$$\bar{R}_i(T) = \mathbb{E} \left[ \sum_{t=1}^{T} (\mu_{i,\bar{m}_i} - X_i(t)) \right],$$

where the expectation is taken over from the randomness of players' strategies and received rewards.

## 4 ALGORITHM

Traditional algorithms typically construct their theoretical frameworks based on the responsive preference assumption. Under this assumption, each arm accepts players according to a fixed preference order, subject to its capacity constraint. This property enables the design of specific player-exploration combinations such that each player in the combination is guaranteed to be accepted by any arm during exploration. Leveraging this, previous algorithms have adopted round-robin exploration mechanisms (Kong & Li, 2024; Zhang & Fang, 2024), ensuring that all exploratory actions yield valid feedback. However, under substitutable preferences, an arm's acceptance decision depends not only on a fixed order but also on the set of applicants, often resulting in only a subset of arms being willing to accept a particular player. As a result, it becomes infeasible to construct exploration combinations that ensure acceptance across all arms. While the existing ODA algorithm (Kong & Li, 2024) attempts to operate under such preference settings, its design essentially sidesteps the core challenge. Specifically, it allows arms to actively select their preferred players, thereby avoiding the incomplete exploration problem rather than directly addressing it and can only guarantee the player-pessimal stable regret. Moreover, these approaches typically assume that arms are endowed with a predefined order, requiring players to explore according to this order—a restriction that is unrealistic in practice. To overcome these limitations, we propose RIFLE—using a randomized active exploration mechanism that allows players to explore arms more effectively under substitutable preferences. This mechanism directly tackles the issue of incomplete feedback caused by rejections during active exploration and, for the first time, guarantees player-optimal stable matchings with the best-known regret bound, even under such a general preference structure. In the following section, we detail RIFLE, which consists of three main phases: (1) an initialization phase for assigning an index to players and arms, (2) a phase for identifying explorable arms, and (3) an explore-then-commit phase.

### 4.1 INITIALIZATION PHASE

The objective of the initialization phase (Algorithm 1) is to assign a unique index to each player who has a chance of being matched and to each arm with a non-empty set of acceptable players.

For clarity, we introduce variables $\text{active}_p[i]$ and $\text{active}_a[j]$ to represent the active state of player i and arm j, respectively. The algorithm consists of two consecutive index assignment phases:

**Player index phase:** This phase assigns index to active players (active$_p[i]$ = True). In each round, the active player uniformly randomly selects one of the $K$ arms to propose (Line 15), and the arm accepts the player based on its preference $\text{Ch}_j(\cdot)$ (Line 16). A player is assigned a unique index only if it is the only one accepted by its selected arm (Line 17– 23). This phase terminates when all players have obtained an index or the maximum number of rounds $T_0$ is reached (Line 13). It is important to note that this phase operates without relying on arm index. This is because the process is driven by players' random proposals, with players only learning from the publicly observable matching results.

**Arm Indexing Phase:** After player indexing is complete, this phase assigns an index to the active arm (active$_a[i]$ = True). In each iteration, all players make uniformly random proposals to the currently active $K_1$ arms (Line 30), and the arms make selections based on their preferences $\text{Ch}_j(\cdot)$ (Line 31). Based on the publicly available matching results, each successfully matched arm j determines the lowest player index among its matches and uses this as the comparison value for this round (Line 35). Each arm then compares these values to determine its relative ranking, obtaining a final index and exiting (Line 36). This phase terminates when all arms have achieved an index or the maximum number of rounds $T_1$ has been reached (Line 28).

Based on this random selection process, it can be shown that after round $T_0$, each player is highly likely to have had at least one opportunity to propose to an arm individually. Therefore, any player whose individual proposal is accepted by at least one arm will receive a unique index during the initialization phase. Conversely, due to the substitutability property, if a player $p_i$ is rejected by arm j even when proposing individually (i.e., $p_i \notin \text{Ch}_j(\{p_i\})$), then that player will also be rejected in any proposal set containing other players. This means that players who fail to obtain an index have no chance of being accepted by any arm and are therefore excluded from the subsequent stages of the algorithm. In the subsequent $T_1$ rounds, since any arm is assumed to find at least one player acceptable, the continued random proposals guarantee that every active arm will eventually accept a player and thereby successfully obtain its own index.

## 4.2 PHASE OF IDENTIFYING EXPLORABLE ARMS

This phase (Algorithm 2) is designed to address the core challenges of exploration under substitutable preferences. Its objective is twofold: first, to systematically identify each player's personalized set of viable arms—those that are willing to accept them. Second, it aims to establish a coordinated, conflict-free exploration mechanism over these identified arms, ensuring that players can gather meaningful feedback efficiently.

Specifically, leveraging the unique indices obtained in Phase 1, players take turns selecting different indexed arms — which we will simply denote as $\{a_1, \ldots, a_K\}$ — in a round-robin fashion (Line 5). If player $p_i$ is accepted by arm $a_j$ when selecting $a_j$ alone, we mark $a_j$ as an explorable arm for $p_i$ and set Explorable$[i][j]$ = True (Line 6). In contrast, for any player–arm pair where Explorable$[i][j]$ = False, according to the substitutability, player $p_i$ will always be rejected by $a_j$ regardless of the presence of other players. Consequently, $p_i$ does not need to explore $a_j$ in the following phases.

After identifying the explorable arms, the algorithm proceeds to determine an exploration mechanism over these arms. For convenience, define $\hat{\mathcal{E}}_i = \{a_j : \text{Explorable}[i][j] = \text{True}\}$ as the set of explorable arms available to player $p_i$. Let $\hat{K}' = \max_i |\hat{\mathcal{E}}_i|$ denote the maximum number of explorable arms among all players. Similarly, define $\hat{N}' = \max_j \sum_i \mathbf{1}\{a_j \in \hat{\mathcal{E}}_i\}$, which represents the maximum number of players that include a particular arm $a_j$ in their explorable set, across all arms. The objective of this sub-phase is to assign explorable arms to players such that, in each round, different players choose different arms, thereby avoiding conflicts. To aid understanding, we first present the core idea of the algorithm in the following paragraph. Lines 10–20 of Algorithm 2 provide a decentralized implementation of this idea.

We adopt a greedy approach in which players select arms based on a predetermined priority index learned in Phase 1. Each player decides whether to select an arm in a given round by observing the selections made by preceding players. Specifically, in round $r$, if none of the preceding players selects arm $a_j$ and player $p_i$ has not selected any other arm, then $a_j$ is available for $p_i$ in that round.

---

**Algorithm 1** Initialization Phase

---

**Require:** Player set $\mathcal{N}$, arm set $\mathcal{K}$, fault-tolerance parameter $\epsilon$

1: **Initialization:**
2:     Proposal probability $p = \frac{1}{K_1}$ for uniform selection, where $K_1$ is active arm number
3:     Player state array $\text{active}_p[i] \leftarrow \text{True}, \forall i \in \mathcal{N}$
4:     Arm state array $\text{active}_a[j] \leftarrow \text{True}, \forall j \in \mathcal{K}$
5:     Player index $\text{index}_p[i] \leftarrow -1, \forall i \in \mathcal{N}$
6:     Arm index $\text{index}_a[j] \leftarrow -1, \forall j \in \mathcal{K}$
7:     Index counter $\text{idx} \leftarrow 0$
8:     Player initialization budget $T_0 = \lceil \frac{\ln(N/\epsilon)}{\ln(1/(1-p(1-p)^{N-1}))} \rceil$
9:     Arm initialization budget $T_1 = K \ln \frac{K}{\epsilon}$
10: **Global Observation:** All players and arms observe the matching results from previous rounds
11: **Player index initialization**
12: **Round counter** $t \leftarrow 0$
13: **while** there exist active players ($\text{idx} < N$) **and** $t < T_0$ **do**
14:     $t \leftarrow t + 1$
15:     Each active player $i$ ($\text{active}_p[i] = \text{True}$) selects arm $A_i(t)$ from $\mathcal{K}$ with probability $p$
16:     Each arm $j$ accepts its most preferred subset based on its own preference $\text{Ch}_j(\cdot)$
17:     **for** arm $j \in \mathcal{K}$ **do**
18:       **if** arm $j$ accepts only one player $p_i$ with $\text{active}_p[i] = \text{True}$ **then**
19:         Assign index: $\text{index}_p[i] \leftarrow \text{idx}$
20:         $\text{active}_p[i] \leftarrow \text{False}$ //Player $i$ exits
21:         $\text{idx} \leftarrow \text{idx} + 1$
22:       **end if**
23:     **end for**
24: **end while**
25: **Arm index initialization**
26: **Round counter** $t \leftarrow 0$
27: **Index counter** $\text{idx} \leftarrow 0$
28: **while** $\text{idx} \leq K$ **and** $t < T_1$ **do**
29:     $t \leftarrow t + 1$
30:     Each player $i$ selects active arm $A_i(t)(\text{active}_a[i] = \text{True})$ from $\mathcal{K}$ with probability $p$
31:     Each arm $j$ accepts its most preferred subset based on its own preference $\text{Ch}_j(\cdot)$
32:     All arms observe the global matching outcome of the current round, $M_t$.
33:     Let $A_t$ be the set of all arms successfully matched in this round, and let $m_t \leftarrow |A_t|$.
34:     **if** arm $j$ is in $A_t$ **then**
35:       Arm $j$ compares value $c_k \leftarrow \min_{p_i \in P_k}\{\text{index}_p[i]\}$ for every arm $k \in A_t$.
36:       Arm $j$ locally sorts all pairs $(c_k, k)$ in ascending order.
37:       Arm $j$ finds its own 0-indexed rank, $r_j$, in the sorted list.
38:       $\text{index}_a[j] \leftarrow \text{idx} + r_j$
39:       $\text{active}_a[j] \leftarrow \text{False}$
40:     **end if**
41:     $\text{idx} \leftarrow \text{idx} + m_t$
42: **end while**
43: **return** Player index $\text{index}_p[i], \forall i \in \mathcal{N}$, Arm index $\text{index}_a[j], \forall j \in \mathcal{K}$

---

It can be shown that within $2 \cdot \max\{\hat{N}', \hat{K}'\}$ rounds, each player can successfully select each of their explorable arms exactly once, with at most one player selecting the same arm in any given round.

### 4.3 EXPLORE-THEN-COMMIT PHASE

The objective of the final phase (Algorithm 3) is to conduct round-robin exploration based on the exploration order learned in Algorithm 2, and to compute the player-optimal stable matching. In general, given the learned exploration order, the explore-then-commit phase follows the core ideas developed in prior works for the one-to-one and responsiveness settings (Zhang et al., 2022; Kong & Li, 2023; 2024). For completeness, we also provide a detailed implementation tailored to the substitutability setting.

---

**Algorithm 2** Phase of Identifying Explorable Arms

---

**Require:** arm set $\mathcal{K}$, the output player index $\text{index}_p[i]$ and arm index $\text{index}_a[j]$ of Algorithm 1
1: **Initialization:** $\text{Explorable}[i][j] = \text{False}$ for all $i \in [N], j \in [K]$
2: // Find the explorable arms for players
3: **for** round $t = 1, 2, ..., \max\{N, K\}$ **do**
4:    **if** $(\text{index}_p[i] + t)\% \max\{N, K\} + 1 \leq K$ **then**
5:       Selects arm $A_i(t) = a_{(\text{index}_p[i]+t)\% \max\{N,K\}+1}$
6:       If the arm accepts $p_i$, update $\text{Explorable}[i][(\text{index}_p[i] + t)\% \max\{N, K\} + 1] = \text{True}$
7:    **end if**
8: **end for**
9: // Determine an exploration order $\left[ E_{i,r_{i\in[N], r\in[2\cdot\max\{\hat{N}', \hat{K}'\}]}} \right]$ over the explorable arms
10: **Initialization:** $E_{i,r} = -1$
11: **for** $\text{idx} = 1, 2, ..., N$ **do**
12:    For convenience, denote $p_i$ as the player with $\text{index}_p[i] = \text{idx}$
13:    **for** $a_j$ in $\hat{\mathcal{E}}_i := \{a_j : \text{Explorable}[i][j] = \text{True}\}$ **do**
14:       **for** $r = 1, 2, ..., 2 \cdot \max\{\hat{N}', \hat{K}'\}$ **do**
15:          **if** $E_{i,r} = -1$ and $\bar{A}_{i'}(r) \neq a_j$ for any $p_{i'}$ with $\text{index}_p[i'] < \text{idx}$ **then**
16:             Selects arm $A_i(t) = a_j$, update $E_{i,r} \leftarrow a_j$
17:          **end if**
18:       **end for**
19:    **end for**
20: **end for**
21: **return** Explorable arms $\hat{\mathcal{E}}_i$ and exploration order $(E_{i,r})_{r\in[2\cdot\max\{\hat{N}', \hat{K}'\}]}, \forall i \in [N]$

---

Algorithm 3 consists of two main parts: exploration (Lines 3–19) and exploitation (Lines 21–25). During the exploration part, each player $p_i$ uniformly explores the arms in $\hat{\mathcal{E}}_i$, following the order $(E_{i,r})_r$ learned from Algorithm 2. The algorithm checks the exploration progress every $2^\ell$ rounds. Once all players have completed their exploration of $\hat{\mathcal{E}}_i$, they proceed to the next part to compute the player-optimal stable matching. In the exploitation part, each player consistently selects their currently estimated most preferred arm. If an arm rejects a player, the player moves on to the next preferred arm in their preference list.

## 5 THEORETICAL RESULTS

We first define some useful quantities in the regret.

**Definition 5.1** (Involved player and arm set size). Define $N' := \max_{j\in[K]} \sum_{i\in[N]} \mathbf{1}\{i \in \text{Ch}_j(\{p_i\})\}$ as the maximum number of players that can be accepted by an arm when proposing it alone. Similarly, define $K' := \max_{i\in[N]} \sum_{j\in[K]} \mathbf{1}\{i \in \text{Ch}_j(\{p_i\})\}$ as the maximum number of arms that would accept a given player when the player proposes alone. It is obvious that $N' \leq N, K' \leq K$.

**Definition 5.2** (Gaps). Define $\Delta_{i,j,j'} = |\mu_{i,j} - \mu_{i,j'}|$ as the preference gap of player $p_i$ over arm $a_j$ and $a_{j'}$. Further, let $\Delta = \min_{i;j,j'\in\{j'':i\in\text{Ch}_{j''}(\{p_i\})\}}$ be the minimum preference gap among all players and their explorable arms.

The following theorem provides the stable regret guarantee for our proposed algorithm.

**Theorem 5.3.** *For any player $p_i$, the player-optimal stable regret over $T$ rounds by following algorithm in Section 4 satisfies*

$$\bar{R}_i(T) \leq O\left(\frac{\max(N', K')\log T}{\Delta^2}\right). \tag{1}$$

Due to the space limit, the proof of Theorem 5.3 is deferred to Appendix B.

Our algorithm introduces a randomized initialization and a novel exploration mechanism to enable systematic and conflict-free exploration in many-to-one matching settings with combinatorially substitutable preferences and eliminating the requirement of predefined arm index. To the best of our

---

**Algorithm 3** Explore-then-commit Phase (from view of $p_i$)

---

**Require:** Player set $\mathcal{N}$, arm set $\mathcal{K}$, time horizon $T$, explore order $(E_{i,r})_r$ from Algorithm 2
1: **Initialization:** empirical mean $\hat{\mu}_{i,j} = 0$, matching counts: $T_{i,j} = 0, \forall a_j \in \mathcal{E}_i$
2: //Round-robin exploration
3: **for** sub-phase $\ell = 1, 2, ...,$ **do**
4:     **for** round $t = 1, 2, ..., 2^\ell$ **do**
5:         Determine the target arm order $r = t\%(2 \cdot \max\{N', K'\})$
6:         Select arm $A_i(t) = E_{i,r}$
7:         **if** $p_i$ is accepted by $A_i(t)$ and receives reward $X_i(t)$ **then**
8:             Update $\hat{\mu}_{i,A_i(t)} \leftarrow (\hat{\mu}_{i,A_i(t)} \cdot T_{i,A_i(t)} + X_i(t))/(T_{i,A_i(t)} + 1), T_{i,A_i(t)} \leftarrow T_{i,A_i(t)} + 1$
9:         **end if**
10:     **end for**
11:     **for** round $t = 1, 2, ..., N$ **do**
12:         **if** $\exists \sigma$ over $\mathcal{E}_i$ such that $\text{LCB}_{i,\sigma_j} > \text{UCB}_{i,\sigma_{j+1}}$ for $j \in [|\mathcal{E}_i| - 1]$ and $\text{index}_p[i] = t$ **then**
13:             Select arm $A_i(t) = a_{\text{index}_p[i]\%K+1}$
14:         **end if**
15:     **end for**
16:     **if** $N'$ players with $\mathcal{E}_i \neq \emptyset$ are all matched in previous $N$ rounds **then**
17:         Enter the next sub-phase
18:     **end if**
19: **end for**
20: //The last sub-phase: finding the player-optimal stable matching
21: Update proposing index $s \leftarrow 1$
22: **for** round $t = 1, 2, ..., T$ **do**
23:     Select arm $A_i(t) = a_{\sigma_s}$
24:     $s \leftarrow s + 1$ if $p_i$ is rejected by $A_i(t)$
25: **end for**

---

knowledge, Theorem 5.3 establishes the first player-optimal regret bound for this general setting. As shown in Table 1, it achieves the optimal regret bounds in previously studied one-to-one and responsive-preference settings, while offering greater generality by removing the need for pre-set arm index and placing no restrictions on the relative sizes of $N$ and $K$.

To demonstrate the effectiveness of our results, we provide experiments in the Appendix C. We compare our approach with existing baselines under substitutable and responsive preferences, demonstrating significant improvements in both matching quality and convergence rate.

# 6   CONCLUSIONS

This paper presents the first player-optimal stable regret analysis for general many-to-one matching markets with substitutable preferences. Our contributions are twofold. First, from the algorithmic perspective, we introduce RIFLE (Randomized Index-Free Learning of Explorable-arms), a general framework that operates under substitutable preference conditions—the most general class considered in matching markets—while avoiding the common assumption of predefined arm indices. Second, from the theoretical perspective, we establish optimal stable regret guarantees: RIFLE achieves the regret bound $O(\max\{N', K'\} \log T/\Delta^2)$, which not only matches the best upper bound in one-to-one markets but also attains the best known upper bound in decentralized many-to-one markets, without imposing restrictions on market size.

Interesting future directions include extending RIFLE to dynamic markets, where the preferences of both players and arms evolve over time. This setting introduces new challenges in tracking and adapting to shifting preferences while maintaining stability and low regret. Another promising direction is the exploration of arms with unknown or partially observable preferences, which would require new learning mechanisms capable of inferring arm-side utilities while preserving stable outcomes.

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

# A    THE USE OF LARGE LANGUAGE MODELS(LLMS)

In writing this article, we used the Large Language Model (LLM) as a writing aid. The use of LLM was limited to improving the text's grammar and accuracy, such as enhancing precise expression, correcting grammar and spelling, and optimizing sentence structure for readability. All core intellectual contributions of this article—including the conceptual framework, research methods, experimental design and analysis, and interpretation of the results—are our original work. LLM was not used to generate hypotheses, conduct analyses, or draw conclusions. We take full responsibility for the intellectual integrity and final content of this article.

# B    PROOF OF THEOREM 5.3

In this section, we will provide the proof of Theorem 5.3.

Recall that our algorithm contains three phases: initialization phase, phase of identifying explorable arms, and the explore-then-commit phase. We will discuss the regret and the properties of each phase separately.

For convenience, we first define the event

$$\mathcal{F}_1 = \{\exists \text{ player } p_i \text{ does not propose any } a_j \text{ alone in any of } T_0 \text{ rounds during Phase 1}\}. \quad (2)$$

$$\mathcal{F}_2 = \{\exists \text{ arm } a_j \text{ does not accept any player } p_i \text{ in any of } T_1 \text{ rounds during Phase 1}\}. \quad (3)$$

## B.1    ANALYSIS OF PHASE 1 (ALGORITHM 1)

**Lemma B.1** (Player Index Assignment Guarantee). *Setting initialization budget*

$$T_0 = \left\lceil \frac{\ln(\frac{NK}{\epsilon})}{\ln\left(\frac{1}{1-p(1-p)^{N-1}}\right)} \right\rceil \quad (4)$$

*with fault-tolerance parameter $\epsilon$ and $p = \frac{1}{K}$, Algorithm 1 guarantees that $\mathbb{P}(\mathcal{F}_1) \leq \epsilon$.*

*In other words, each player $p_i$ such that $\exists j \in [K]$ with $p_i \in \text{Ch}_j(\{p_i\})$ can obtain a unique index with probability at least $1 - \epsilon$.*

*Proof.* Recall that each player (with no index yet) will select each arm with probability $p = 1/K$. Fix an arm $a_j$, then the probability that $p_i$ selects $a_j$ alone is $p(1-p)^{N-1}$. The probability that $p_i$ does not select any arm alone over the $T_0$ rounds can be bounded by

$$\begin{aligned}
\mathbb{P}(\mathcal{F}_1) =& \mathbb{P}\big(\exists \text{ player } p_i \text{ does not propose any } a_j \text{ alone in any of } T_0 \text{ rounds}\big) \\
\leq& \sum_{i \in [N]} \mathbb{P}(p_i \text{ does not propose } a_j \text{ alone in round } t, \forall j \in [K], t \in [T_0]) \\
\leq& \sum_{i \in [N]} \sum_{j \in [K]} \mathbb{P}(p_i \text{ does not propose } a_j \text{ alone in round } t, \forall t \in [T_0]) \\
\leq& NK\left(1 - p(1-p)^{N-1}\right)^{T_0} \\
\leq& \epsilon,
\end{aligned}$$

where the last inequality holds based on the selection of $T_0$.

Therefore, with probability larger than $1-\epsilon$, each player will have a chance to select each arm alone. And all players will obtain a unique index if $\exists j$ such that $p_i \in \text{Ch}_j(\{p_i\})$. □

**Lemma B.2** (Complexity of $T_0$). *The selection of $T_0$ in Eq.equation 4 satisfies*

$$T_0 \leq K(\ln NK + \ln 1/\epsilon).$$

*Proof.* Define function $h(x) = \ln(1 - x) + x$, where $x \in (0, 1)$. The derivative of $h(x)$ satisfies $h'(x) = 1/(x - 1) + 1 < 0$. So $h(x)$ is decreasing on $(0, 1)$ and $h(x) < h(0) = 0$, which means $x \leq -\ln(1 - x)$.

Letting $x = p(1 - p)^{N-1} = 1/K \left(1 - 1/K\right)^{N-1}$, then

$$T_0 \leq \frac{\ln NK - \ln \epsilon}{-\ln\left(1 - \frac{1}{K}(1 - \frac{1}{K})^{N-1}\right)} + 1 \leq \frac{\ln NK - \ln \epsilon}{\frac{1}{K}(1 - \frac{1}{K})^{N-1}} + 1 = \frac{K(\ln NK - \ln \epsilon)}{(1 - \frac{1}{K})^{N-1}} + 1 \,.$$

Further, define $g(x) = (1 - x) - \exp\left[\frac{-x}{1-x}\right]$ for $x \in (0, 1)$. Take the derivative of $g(x)$, we have $g'(x) = -1 + \frac{1}{(x-1)^2}\exp\left[1 + \frac{1}{x-1}\right] > 0$. So $g(x) > g(0) = 0$.

By setting $x = 1 - 1/K$, we have

$$\left(1 - \frac{1}{K}\right)^{N-1} > \left(1 - \frac{1}{K}\right)^{-1} e^{-\frac{\frac{N}{K}}{1 - \frac{1}{K}}} = \left(1 - \frac{1}{K}\right)^{-1} e^{-\frac{N}{K-1}} \,.$$

In summary, $T_0$ can be scaled to

$$T_0 \leq \frac{K(\ln NK - \ln \epsilon)}{\left(1 - \frac{1}{K}\right)^{N-1}} + 1 \leq \frac{K \ln (NK/\epsilon)}{\left(1 - \frac{1}{K}\right)^{-1} e^{-\frac{N}{K-1}}} + 1$$

$$\leq \frac{(K-1) \ln (NK/\epsilon)}{e^{-\frac{N}{K-1}}} + 1 = O\left(K \ln NK + \ln \frac{1}{\epsilon}\right) \,.$$

$\square$

**Lemma B.3** (Complexity of $T_1$). *Under the error tolerance of $\epsilon$, the upper bound of the total number of rounds required for Arm initialization is*

$$T_1 = K ln \frac{K}{\epsilon} \tag{5}$$

*In other words, each arm $a_{ij}$ can obtain a unique index with probability at least $1 - \epsilon$.*

*Proof.* We assume $K_{min}$ is the minimum capacity of all arms, and $m$ is the total number of arms that have not yet exited the match in each round. For any arm j, let $A_j$ be the total number of players it can accept. In any round, the probability that $arm_j$ is not proposed by a player that it can accept is

$$p_j = \left(1 - \frac{1}{m}\right)^{A_j}.$$

After repeating $T_j$ rounds independently, the probability that $arm_j$ is not hit in all $T_j$ rounds does not exceed

$$\mathbb{P}\left(\mathcal{F}_2\right) = p_i^{T_j} = \left(1 - \frac{1}{m}\right)^{A_j T_j}.$$

To ensure that the failure probability of each branch is at most $\epsilon/K$, we only need to set

$$\left(1 - \frac{1}{m}\right)^{A_j T_j} \leq \frac{\epsilon}{K}.$$

Taking the logarithm of both sides and noting that $\ln(1 - \frac{1}{m}) < 0$, we obtain

$$A_j T_j \ln\left(1 - \frac{1}{m}\right) \leq \ln\left(\frac{\epsilon}{K}\right) \implies T_j \geq \frac{\ln(K/\epsilon)}{-A_j \ln(1 - \frac{1}{m})}.$$

Next, we need to find the maximum value on the right side such that $T_j$ satisfies the condition regardless of the arm. First, we lower-bound the capacity using $A_j \geq K_{min}$ and then set $m$ to $K$, obtaining

$$T_1 \geq \frac{\ln(K/\epsilon)}{-K_{min} \ln(1 - \frac{1}{K})}.$$

Using the standard inequality $\ln(1 + t) \le t$ (where $t > -1$), setting $t = -\frac{1}{K}$ yields

$$-\ln\left(1 - \tfrac{1}{K}\right) \ge \tfrac{1}{K},$$

Thus,

$$\frac{\ln(K/\epsilon)}{K_{min} \cdot (1/K)} = \frac{K}{K_{min}} \ln\frac{K}{\epsilon} \le T_1.$$

Since we know that each arm will accept at least one player in a match, we have $K_{min} \ge 1$. Thus

$$\frac{K}{K_{min}} \ln\frac{K}{\epsilon} \le K \ln\frac{K}{\epsilon} \le T_1.$$

So when $T_1 \ge K \ln\frac{K}{\epsilon}$, it is guaranteed that for each arm $j$, $\mathbb{P}(\mathcal{F}_2) \le \epsilon/K$ exists. That is, with a probability of at least $1 - \epsilon$, all arms are successfully indexed within $T_1$ rounds. The proposition is proved. $\qquad\square$

## B.2 ANALYSIS OF PHASE 2 (ALGORITHM 2)

**Lemma B.4** (Identity of Explorable Arms). *Define* $\mathcal{E}_i = \{a_j : p_i \in \text{Ch}_j(\{p_i\})\}$. *Conditional on event* $\neg\mathcal{F}_1$, *the output* $\hat{\mathcal{E}}_i$ *of Algorithm 2 satisfies* $\hat{\mathcal{E}}_i = \mathcal{E}_i$. *Further,* $\hat{N}' = N', \hat{K}' = K'$.

*Proof.* Conditional on $\neg\mathcal{F}_1$, each player $p_i$ with index$[i] \ne -1$ has a unique index different from others. So in the first $\max\{N, K\}$ rounds of Algorithm 2, each player selects different arms in each round. If $p_i \in \text{Ch}_j(\{p_i\})$, then $p_i$ will mark Explorable$[i][j]$ as True in the round of selecting $a_j$. Thus $\hat{\mathcal{E}}_i = \mathcal{E}_i$. Based on the definition, it is obvious that $\hat{N}' = N', \hat{K}' = K'$. $\qquad\square$

**Lemma B.5** (Exploration Order). *Conditional on* $\neg\mathcal{F}_1$ *and based on Lemma B.4, the output* $(E_{i,r})_{i \in [N], r \in [2 \max\{N', K'\}]}$ *of Algorithm 2 satisfies*

*(1)* $\forall r, \forall i \ne i', E_{i,r} \ne E_{i',r}$ *when both* $E_{i,r} \ne -1$ *and* $E_{i',r} \ne -1$.

*(2)* $\forall i, \mathcal{E}_i \subseteq \cup_r E_{i,r}$.

*Proof.* Condition (1) holds obviously based on Line 15 of Algorithm 2. In the following, we will prove condition (2).

By contradiction, suppose in $2 \max\{N', K'\}$ rounds, $\exists p_i, a_j$ such that $a_j \in \mathcal{E}_i$ but $a_j \notin \cup_r E_{i,r}$. Let $p_i$ and $a_j$ be the first such pair during Line 10-20. Then it means that when $p_i$ update $E_{i,r}$, no $r \in [2 \cdot \max\{N', K'\}]$ satisfies that $E_{i,r} = -1$ and $E_{i',r} \ne a_j$.

For previous players $p_{i'}$ with index$[i'] < $ index$[i]$, they select arm $a_j$ in at most $N'$ rounds. That means $|\cup_r\{\exists i', E_{i',r} = a_j\}| \le N'$. And for player $p_i$, it selects other arms in $\mathcal{E}_i$ in at most $K'$ rounds, which means $|\cup_r\{E_{i,r} \ne -1\}| \le K'$. So there must exists available $r \in [2 \cdot \max\{N', K'\}]$ such that $E_{i,r} = -1$ and $E_{i',r} \ne a_j$. This introduces a contradiction. $\qquad\square$

## B.3 ANALYSIS OF PHASE 3 (ALGORITHM 3)

Define event

$$\mathcal{F}_3 = \left\{ \exists t \in [T], i \in [N], j \in [K] : |\hat{\mu}_{i,j}(t) - \mu_{i,j}| > \sqrt{\frac{6 \log T}{T_{i,j}(t)}} \right\}.$$

to represent that the estimated preference value is far from the real preference value in some round.

The following lemma bounds the probability of this event.

**Lemma B.6** (Probability of The Failure Event).

$$T \cdot \mathbb{P}(\mathcal{F}_3) \le 2NK.$$

*Proof.*

$$T \cdot \mathbb{P}(\mathcal{F}_3) = T \cdot \mathbb{P}\left(\exists t \in [T], \ i \in [N], \ j \in [K] : |\hat{\mu}_{i,j}(t) - \mu_{i,j}| > \sqrt{\frac{6 \log T}{T_{i,j}(t)}}\right)$$

$$\leq T \cdot \sum_{t=1}^{T} \sum_{i \in [N], j \in [K]} \mathbb{P}\left(|\hat{\mu}_{i,j}(t) - \mu_{i,j}| > \sqrt{\frac{6 \log T}{T_{i,j}(t)}}\right)$$

$$\leq T \cdot \sum_{t=1}^{T} \sum_{i \in [N], j \in [K]} \sum_{w=1}^{t} \mathbb{P}\left(T_{i,j}(t) = w, \quad |\hat{\mu}_{i,j}(t) - \mu_{i,j}| > \sqrt{\frac{6 \log T}{T_{i,j}(t)}}\right)$$

$$\leq T \cdot \sum_{t=1}^{T} \sum_{i \in [N], j \in [K]} t \cdot 2 \exp(-3 \log T)$$

$$= T \cdot \sum_{t=1}^{T} \sum_{i \in [N], j \in [K]} t \cdot \frac{2}{T^3}$$

$$\leq 2NK .$$

The third inequality comes from Lattimore & Szepesvári (2020, Corollary 5.5). □

**Lemma B.7** (Complexity of The First Sub-phase in Algorithm 3). *Conditional on $\neg \mathcal{F}_3$, Line 17 of Algorithm 3 happens at phase $\ell_{\max}$ which satisfies*

$$\ell_{\max} = \min\{\ell : \sum_{\ell=1}^{\ell_{\max}} 2^\ell \geq 192 \max\{K', N'\} \log T / \Delta^2\} .$$

*Proof.* By contradiction, suppose there exists player $p_i$ and arm $a_j, a_{j'} \in \mathcal{E}_i$ with $\mu_{i,j} < \mu_{i,j'}$ such that $\mathrm{LCB}_{i,j'} \leq \mathrm{UCB}_{i,j}$. It implies that

$$\mu_{i,j'} - 2\sqrt{\frac{6 \log T}{T_{i,j'}(t)}} \ \leq \ \mathrm{LCB}_{i,j'}(t) \ \leq \ \mathrm{UCB}_{i,j}(t) \ \leq \ \mu_{i,j} + 2\sqrt{\frac{6 \log T}{T_{i,j}(t)}} .$$

It follows that

$$\Delta_{i,j,j'} = \mu_{i,j'} - \mu_{i,j} \ \leq \ 4\sqrt{\frac{6 \log T}{\min\{T_{i,j}(t), \ T_{i,j'}(t)\}}} . \tag{6}$$

However, based on the algorithmic procedure, in every $2 \cdot \max\{N', K'\}$ rounds, each player will match each explorable arm at least once. So at the end of the phase $\ell_{\max}$, it holds that

$$\min\{T_{i,j}(t), \ T_{i,j'}(t)\} \geq 96 \log T / \Delta^2 .$$

This leads to a contradiction with Eq.equation 6. □

**Lemma B.8** (Estimation Correctness). *Conditional on $\neg \mathcal{F}_3$, $\mathrm{UCB}_{i,j} < \mathrm{LCB}_{i,j'}$ implies that $\mu_{i,j} < \mu_{i,j'}$. So the estimated preference ranking $\sigma$ in the first sub-phase is correct.*

*Proof.* By the definition of UCB and LCB as well as event $\neg \mathcal{F}_3$, it is obvious that

$$\mu_{i,j} \leq \mathrm{UCB}_{i,j} < \mathrm{LCB}_{i,j'} \leq \mu_{i,j'} .$$

□

### B.4 PROOF OF THEOREM 5.3

Based on the analysis of three phases, now we are ready to bound the total regret. Let $T_{sum} = T_0 + T_1$ is the index of the final round of Phase 1. For convenience, denote $T_2$ as the final round index of Phase 2. The the regret satisfies

$$\bar{R}_i(T) = \mathbb{E}\left[\sum_{t=1}^{T}\left(\mu_{i,\bar{m}_i} - X_i(t)\right)\right]$$

$$\leq \mathbb{E}\left[\sum_{t=1}^{T_{sum}}\left[\mu_{i,\bar{m}_i} - X_i(t)\right]\right] + \mathbb{E}\left[\sum_{t=T_{sum}+1}^{T_2}\left[\mu_{i,\bar{m}_i} - X_i(t)\right]\right] + \mathbb{E}\left[\sum_{t=T_2+1}^{T}\left[\mu_{i,\bar{m}_i} - X_i(t)\right]\right]$$

$$\leq T_{sum}\mu_{i,\bar{m}_i} + \left(\max\{N,K\} + NK' \cdot 2\max\{N',K'\}\right)\mu_{i,\bar{m}_i} + \mathbb{E}\left[\sum_{t=T_2+1}^{T}\left(\mu_{i,\bar{m}_i} - X_i(t)\right)\right]$$

$$\tag{7}$$

where the last inequality using the length of $T_2 - T_{sum}$ is based on the procedure of Algorithm 2. In the following, we mainly aim to bound the third term in Eq.equation 7.

$$\mathbb{E}\left[\sum_{t=T_2+1}^{T}\left(\mu_{i,\bar{m}_i} - X_i(t)\right)\right]$$

$$\leq \mathbb{E}\left[\sum_{t=T_2+1}^{T}\left(\mu_{i,\bar{m}_i} - X_i(t)\right) \mid \neg\mathcal{F}_1 \cap \neg\mathcal{F}_2 \cap \neg\mathcal{F}_3\right] + T\left[\mathbb{P}\left(\mathcal{F}_1\right) + \mathbb{P}\left(\mathcal{F}_2\right) + \mathbb{P}\left(\mathcal{F}_3\right)\right]\mu_{i,\bar{m}_i}$$

$$\leq \mathbb{E}\left[\sum_{t=T_2+1}^{T}\left(\mu_{i,\bar{m}_i} - X_i(t)\right) \mid \neg\mathcal{F}_1 \cap \neg\mathcal{F}_2 \cap \neg\mathcal{F}_3\right] + 2\mu_{i,\bar{m}_i} + 2NK\mu_{i,\bar{m}_i} \tag{8}$$

$$\leq \mathbb{E}\left[\sum_{\ell=1}^{\ell_{\max}}\left(2^\ell + N\right) + NK\right] \cdot \mu_{i,\bar{m}_i} + (2 + 2NK)\mu_{i,\bar{m}_i} \tag{9}$$

$$\leq 192\max\{N',K'\}\log T/\Delta^2 + N\log\left(192\max\{N',K'\}\log T/\Delta^2\right) + NK\mu_{i,\bar{m}_i}$$
$$+ (2 + 2NK)\mu_{i,\bar{m}_i}. \tag{10}$$

where Eq.equation 8 holds based on Lemma B.1 by selecting $\epsilon = 1/T$ and Lemma B.6. Eq.equation 9 is because, based on Lemma B.8, the procedure of the second sub-phase of Algorithm 3 equals to the offline Gale-Shapley algorithm. Since each arm can reject each player at most once, so the time complexity of the second sub-phase before reaching player-optimal stability is $NK$. Eq.equation 10 is based on the value of $\ell_{\max}$ in Lemma B.7.

Finally, combining Eq.equation 10 with Eq.equation 7, as well as the complexity $T_0$ in Lemma B.2 with $\epsilon = 1/T$, we can obtain that

$$\bar{R}_i(T) \leq O(\max\{N',K'\}\log T/\Delta^2).$$

## C  EXPERIMENTS

In this section, we compare our proposed algorithm with existing methods under different many-to-one matching models. We evaluate the performance of algorithms in terms of both stable regret and market unstability, which represents the number of cumulative unstable matchings.

We designed three experiments to systematically evaluate the performance of different algorithms under substitutable and responsive preferences. The first two experiments are based on substitutable preferences: the first examines the difference in optimal regret when more than one stable match exists; the second compares the difference in optimal regret between ODA and our method when only one stable match exists. For both experiments, we selected preference gap parameters $\Delta \in \{0.3, 0.4, 0.5\}$ for testing; however, due to the different properties of the two preference structures, to ensure that the algorithms have sufficient rounds to demonstrate long-term behavior in their respective environments, the number of rounds for the former was set to 60k, and for the latter to 200k. The third experiment is based on responsive preference, comparing the performance of ODA, AETDA, and our method under optimal regret. For ease of illustration and comparison, this experiment uses a representative gap $\Delta = 0.5$ and a total of 40k rounds. The final result is the average of 5 independent runs to reduce the impact of randomness and highlight performance differences.

For the first experiment, in this setting, to the best of our knowledge, the only available baseline is the ODA algorithm (Kong & Li, 2024). We first compare their performances in terms of the player-optimal stable matching, i.e., both the stable regret and instability are computed compared with the player-optimal stable matching. The results are shown in Figure 1.

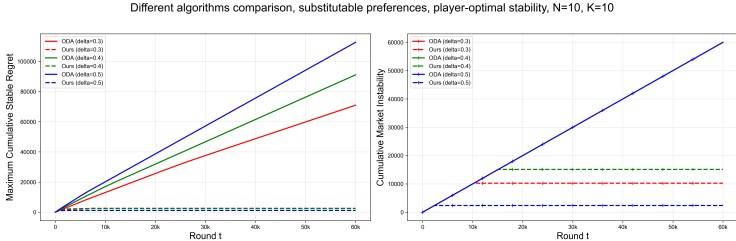

Figure 1: Comparison between our algorithm and ODA in terms of player-optimal stability.

As shown in the figure, our algorithm (dashed) consistently outperforms ODA (solid) across all $\Delta$ settings. ODA's regret and instability fail to converge within the time horizon, consistent with its guarantee of only reaching player-pessimal stability. As expected, smaller $\Delta$ leads to higher regret.

To further evaluate matching efficiency beyond matching quality, we report the stable regret and instability of the ODA algorithm under the player-pessimal stable matching. Specifically, we report the performance of our algorithm in terms of the player-optimal stable matching and that of ODA in terms of the player-pessimal stable matching. The results are shown in Figure 2.

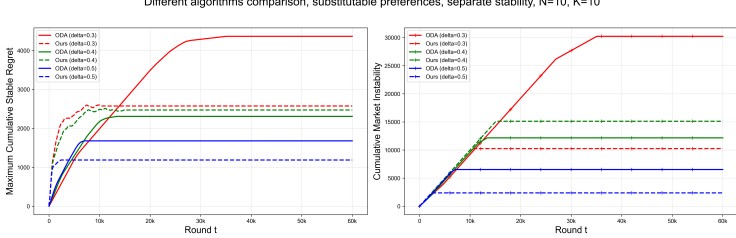

Figure 2: Comparison between our algorithm and ODA in terms of separate stability.

As illustrated in the figures, our algorithm not only guarantees convergence to the player-optimal stable matching but also achieves higher learning efficiency compared with ODA (Kong & Li, 2024), which can only converge to the player-pessimal stable matching. In particular, while Figure 1 shows that ODA suffers from ever-increasing regret and instability, Figure 2 demonstrates that even when both algorithms are evaluated purely in terms of stable regret under their respective convergence guarantees, our algorithm consistently attains significantly lower regret.

In the second experiment, we only considered the case of a unique stable match under substitutable preference (see Figure 3). The experimental results show that although both ODA and our method eventually converge to a stable match, there is a significant difference in the speed at which they reach stability: our algorithm converges to a stable match much faster, while ODA continues to explore for a longer period. Furthermore, the cumulative regret curve also shows that our method has significantly lower cumulative regret before and after reaching stability, indicating that it is more efficient in learning arm preferences and converging to stable matches. This phenomenon is consistent with our theoretical analysis, namely that in environments with multiple stable matches, our method can identify the optimal stable match faster while avoiding unnecessary exploration, thus achieving lower stability regret.

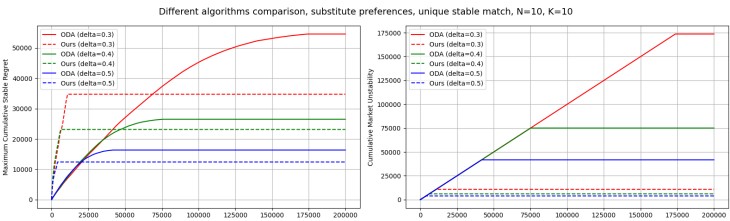

Figure 3: Comparison between our algorithm and ODA under unique stable match.

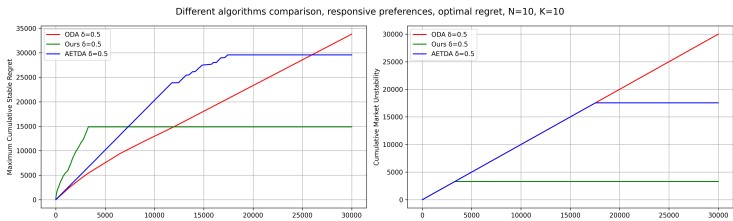

Figure 4: Comparison between our algorithm, ODA and AETDA under responsive preference.

In the third experiment, we further examined the optimal stable regret performance of each algorithm under the responsive preferences condition. Since Zhang & Fang (2024)'s method is a centralized setting, and the original paper does not provide a fully reproducible algorithm flow, Wang et al. (2022)'s optimal-regret method relies on the assumption that it $\Delta$ is known, which is inconsistent with our focus on online learning scenarios; this experiment only compares it with AETDA (Kong & Li, 2024). The experimental results are shown in Figure 4.

Under this setting, we observed that ODA failed to converge, which is consistent with its design goal of optimizing pessimal regret rather than optimal regret, thus it cannot approximate the true optimal stable match. In contrast, both AETDA and our method achieved convergence; more importantly, our method converged significantly faster and had a significantly smaller cumulative stable regret, demonstrating better learning efficiency and stability. This experiment is consistent with the observations in the previous two experiments, further validating the effectiveness and advantages of our method within the responsive preference framework.

