# OpenReview forum: "Player-optimal Stable Regret for Bandit Learning in Many-to-one Matching Markets with Substitutability"
_ICLR.cc/2026/Conference — Submitted to ICLR 2026_

### Official Review · Reviewer_aqJY · 2025-10-21

**Soundness:** 3
**Presentation:** 3
**Contribution:** 2
**Rating:** 4
**Confidence:** 4

**Summary:**

The paper studies the many-to-one matching markets under bandit framework, focusing on substitutability assumption. It assumes that the players (the one side) have unknown preferences while arms (the many side) have known preferences. The paper aims to find the player-optimal stable matching, and proposes an algorithm that consists of three phases: initialization, indentifying explorable arms, and explore-then-commit. The logarithmic cumulative regret is established.

**Strengths:**

1. The paper provides extensive comparison with existing literature.

2. The theoretical claim looks solid, and simulated experiments are conducted to complement the theory.

**Weaknesses:**

1. The algorithmic presentation needs improvement. What is the key difference of substitutability compared to responsiveness that makes the problem more difficult? Can you give an example with substitutability such that AETDA (Kong & Li 2024) failed to identify the player-optimal stable matching? What is the key novelty of the proposed algorithm that solved the problem?

2. The technical contribution is rather limited. Compared to Kong 2024, the only contribution of this work is that the paper studies the player-optimal stable regret with substitutability and there is a $1/N$ improvement. The algorithm also looks similar to theirs, all utilizng the UCB structure to eliminate arms.

**Questions:**

1. Please see the weakness part. The paper presentation might benefit from examples, especially to readers that are not familiar with the line of work.

2. I suggest the authors to name the algorithm.

3. For the experiments, have you considered implementing algorithms when there only exists a unique matching especially when your baseline algorithm is evaluated on player-pessimal stable regret?

4. What is the lower regret bound of the problem?

---

> ### Author Response · Authors · 2025-11-23
>
> Thank you for your careful reading and constructive comments. Below are responses to your main questions and suggestions.
>
> First, regarding the complexity of substitutable preferences compared to responsive ones and the distinction from AETDA (Kong \& Li 2024). Intuitively, responsive preferences ensure monotonic consistency based on a linear ordering, which AETDA mainly relies on to infer explorable sets. In contrast, substitutable preferences depend on combinatorial structures. For example, consider a supplier with a capacity of 2 facing brands $\{H_1,H_2,L_1,L_2\}$. Its selection rule is to prioritize diversity: it selects one high-end ($H$) and one ordinary ($L$) brand if both types are available; otherwise, it selects the top two from the single available category. While this satisfies substitutability, it cannot be induced by any linear ordering (e.g., preferring $\{H_1, L_1\}$ over $\{H_1, H_2\}$ even if $H_2$ ranks higher than $L_1$ individually), thus violating responsiveness. Consequently, players proposing simultaneously might be rejected due to the specific combination rather than individual unacceptability. This leads existing methods to incorrectly narrow the explorable set, preventing convergence to a player-optimal stable matching. While ODA (Kong \& Li 2024) addresses substitutability, it relies on arm-side proposals, rendering players passive and making player-optimality difficult to guarantee. To address this, our algorithm is designed from the player's perspective: we explicitly construct a personalized set of explorable arms via a finite, conflict-free exploration sequence. This ensures all potential matches are discovered, achieving player-optimal stable regret under general substitutability.
>
> Secondly, regarding the concern that our contribution merely extends Kong \& Li (2024) with minor $1/N$ improvements or relies solely on a UCB structure, we highlight three key innovations: 1) General model. We move from responsive to fully substitutable many-to-one preferences and give the first player-optimal stable regret analysis under this general setting. 2) Index-free decentralization. Unlike prior works that assume a globally known arm index, our method starts with no index and lets each player infer a consistent arm order purely from matching feedback, enabling decentralized learning under substitutable preferences. 3) Optimal regret. Under these weaker assumptions, our regret $O(\max(N,K)\log T/\Delta^2)$ matches the best known order in one-to-one and responsive markets, far beyond a small $1/N$ refinement.
>
> Regarding the UCB structure, we agree with the reviewers' observation: both we and Kong & Li 2024 adopted a UCB structure, but the UCB's role is limited to helping the player distinguish the superiority and inferiority of each arm on a given exploration sequence. The main innovation of this paper lies in how, under a substitutable + no pre-defined index setting, we, through the design of Phase 1 and 2, enable players to find a conflict-free and sufficient exploration sequence with the smallest possible regret, and then superimpose UCB on that sequence for preference learning.
>
> We also strongly agree with your suggestions regarding demonstrability and experimentation and have improved upon them in the revised manuscript. Specifically: (1) We have named the main algorithm for use in the paper (2) In the experimental section, in addition to the comparison with ODA, we supplemented the performance in the case of a unique stable matching. In this case of a unique stable matching, the difference between player-optimal and player-pessimal disappears, and the methods converge in the final matching quality. However, our algorithm still has good competitiveness in terms of convergence speed and regret, thus more comprehensively demonstrating the advantages and limitations of our method.
>
> Finally, regarding the lower bound of regret. In the current work, we have not given a new lower bound result for formally complete matching under the general substitutable many-to-one setting, which remains an open question in theory. However, existing literature provides strong references for simpler one-to-one cases: Sankararaman, Basu, and Sankararaman (2021) simultaneously give the upper bound of $O(NK\log T/\Delta^2)$ and the lower bound of $\Omega(N\log T/\Delta^2)$ within the one-to-one (serial dictatorship) framework, clearly showing that in the matching-bandit problem, the dependency on $\log T$ and $\Delta^{-2}$ is already a fundamental bottleneck in information theory.

---

> > ### Author Response · Authors · 2025-12-02
> >
> > Since we did not receive further follow-up from the reviewer during the discussion phase, we have made targeted improvements addressing the reviewer’s main concerns on problem difficulty/novelty, presentation, and experimental completeness. We also incorporated the key explanations and newly added content into the revised manuscript and experiments to facilitate the AC’s independent evaluation. The main updates are as follows:
> >
> > **1. Key differences between substitutable and responsive preferences:** In the rebuttal, we provided a counterexample that satisfies substitutability but violates responsiveness, showing that algorithms relying on a linear ranking structure can fail. This motivates the necessity of our conflict-avoidance mechanism.
> >
> > **2. Clearer distinction from Kong & Li 2024 (ODA/AETDA):** Our core contribution is not simply adopting UCB. Instead, under the weaker information structure of **substitutable preferences with no pre-defined arm index**, we design Phases 1–2 to construct a conflict-free exploration schedule that supports learning the player-optimal stable regret; UCB is **only used to distinguish preferences along this schedule**.
> >
> > **3. Improved readability and naming the algorithm:** Following the reviewer’s suggestion, we have named the main algorithm in the paper to make it easier to reference and understand.
> >
> > **4. More comprehensive experimental settings:** In the revised version, we added experiments under **a unique stable matching setting**. This shows that when the difference between player-optimal and player-pessimal matching outcomes disappears, our method still **maintains strong convergence speed and regret performance**, providing a more complete evaluation of strengths and limitations.
> >
> > **5. On the lower bound:** While we do not provide a new lower bound for the general substitutable many-to-one setting, we added references to known information-theoretic lower bounds in simpler one-to-one cases to justify the unavoidable dependence on $\log T$ and $\Delta^{-2}$, and clarified that our upper bound matches the best-known order.

---

### Official Review · Reviewer_QDsS · 2025-10-30

**Soundness:** 3
**Presentation:** 2
**Contribution:** 2
**Rating:** 4
**Confidence:** 3

**Summary:**

This paper studies the problem of bandit learning in many-to-one matching markets under the general substitutability condition. While previous works have mostly focused on one-to-one settings or many-to-one settings with responsive preferences, this work addresses the more challenging scenario where arms (e.g., resources or positions) have substitutable preferences over sets of players. The authors propose a novel algorithm that combines randomized initialization and a decentralized, conflict-free exploration mechanism, enabling players to efficiently identify explorable arms and ultimately converge to the player-optimal stable matching. Theoretical analysis shows that the proposed method achieves a regret bound of $O(\max{K,N}\log T/\Delta^2)$, which matches or improves upon the best known results in both one-to-one and many-to-one settings. Empirical results further demonstrate the superiority of the proposed approach over existing baselines in terms of both matching quality and convergence speed.

**Strengths:**

1. The paper addresses an important and challenging problem of bandit learning in many-to-one matching markets with substitutable preferences, representing a significant generalization over prior works that mainly consider responsive preferences or one-to-one settings.

2. The regret analysis is rigorous, and the theoretical results are strong, establishing the first player-optimal stable regret bound under substitutable preferences.

3. The paper is generally well-written, with clear and detailed explanations of the problem formulation, algorithmic design, and theoretical analysis. The related work section is comprehensive and provides a good context for the contributions.

**Weaknesses:**

1. The main innovation lies in the initialization phase (using a musical chairs approach) and the exploration strategy for identifying active arms. However, these techniques may not be particularly surprising or fundamentally novel.

2. While the paper claims to be fully decentralized with no predetermined indices for players or arms, it assumes that the matching outcome is globally observed by all players. This assumption may not be realistic in practical decentralized systems and could limit the applicability of the approach.

3. The experimental evaluation is mainly limited to comparison with the ODA algorithm. Including more baselines, such as algorithms designed for responsive preferences, would provide a more comprehensive assessment of the proposed method’s advantages and limitations.

**Questions:**

1. Will players share an identical index?

---

> ### Author Response · Authors · 2025-11-23
>
> Thank you for your detailed comments on our work. Regarding the points you raised, we offer the following responses and clarifications:
>
> First, our approach differs fundamentally from the classic musical chairs mechanism in both its setup and objectives. Traditional musical chairs typically implies a one-to-one scenario: each arm can only accept at most one player at a time. Through a random proposal process, a unique matching arm is found for each player, eliminating those who cannot take a seat. In our setup, however, each arm has capacity and can accept multiple players simultaneously. Therefore, the analytical structure of classic musical chairs does not directly apply here. Furthermore, under the substitution preference framework, there might be situations where an arm does not accept a particular player combination but accepts only one player from that combination. If the musical chairs approach is used simply, an arm could match multiple players, making it impossible to distinguish between different players. In the worst case, this could result in completely indistinguishable player combinations, preventing players from fully exploring their explorable combinations. Therefore, in Phase 1, our goal is not to eliminate players, but rather to differentiate players and assign them independent indices through multiple rounds of random matching, while allowing an arm to accept multiple players simultaneously. In subsequent steps, the index of each arm are also gradually determined using similar rules based on matching results. The resulting index provide the necessary foundation for decentralized scheduling and conflict-free exploration in Phases 2 and 3, thus supporting subsequent player-optimal stable regret analysis.
>
> Secondly, regarding the question of whether players need to share the same index in a fully decentralized environment where publicly observable matching results exist, our assumption is that at the end of each round, the current market matching results are visible to all players, and each player only observes their own reward. In Phase 1, as long as a player is uniquely accepted by a particular arm in a round, they can independently infer their own and each arm's index using round t and historical matching data. Since the rules are public and deterministic, as long as all players see the same matching history, they will logically obtain the same set of indexes without any communication.
>
> The coordination in Phase 2 also relies entirely on public rules and matching feedback, and can be considered a communication-free but stable implicit coordination mechanism that does not require central control or information exchange between players. We prove that this mechanism only introduces a polynomial factor related to $\max(N',K')$, while maintaining a $\log T$ level in the time dimension, thus ensuring that the overall regret is quadratic.
>
> It is worth emphasizing that this assumption of observable matching results but no communication has been adopted in a large number of decentralized matching learning papers. For example, Liu et al. (2021) uses the matching results of arms to construct conflict information to reduce invalid conflicts, and Kong & Li (2023) and Kong et al. (2024) also achieve player synchronization in a communication-free environment by observing the final matching of arms in each round. Therefore, our setting is completely consistent with and reasonable with existing research.
>
> Finally, regarding the choice of experimental baseline, we agree that a comparison with ODA alone is insufficient. Therefore, in this revised version, we supplement the work with algorithms known to be best performing under responsive preferences. Considering that Zhang & Fang (2024)'s method is centralized and does not provide a complete reproducible implementation in the paper, and Wang et al. (2022)'s optimal regret method relies on the known preference gap parameter $\Delta$, which is not suitable for our online learning scenario, this experiment only adds AETDA (Kong & Li, 2024). A systematic comparison with our method and ODA is then conducted to more clearly demonstrate that we can achieve player-optimal regret not only under substantial preference but also consistent with the current best regret under responsible preference. We believe these additions will more strongly support the theoretical and experimental contributions of this paper.

---

> > ### Author Response · Authors · 2025-12-02
> >
> > Since we did not receive further follow-up during the discussion phase and the reviewer’s main concerns centered on algorithmic novelty and the realism of our assumptions, we summarize our key responses below. These clarifications and additions have been incorporated into the revised manuscript to support the AC’s independent evaluation.
> >
> >  **1. Clarifying the distinction from “Musical Chairs” and highlighting novelty:** Classical musical chairs mainly applies to **one-to-one settings**. In many-to-one markets with substitutable preferences, acceptance depends on the proposal set, **making feedback inherently ambiguous and a direct adaptation ineffective**. Phase 1 is specifically designed to resolve this set-dependent ambiguity by constructing a consistent index via repeated random matching, enabling subsequent decentralized, conflict-free exploration.
> >
> >  **2. Global observability assumption:**  Our assumption only requires that each round’s matching **outcome is publicly observable; players observe only their own rewards and do not communicate**. This supports implicit coordination rather than centralized assignment and aligns with standard assumptions in prior decentralized matching learning work.
> >
> >  **3. Adding additional baselines under responsive preferences:** The revision adds a representative **SOTA baseline under responsive preferences (AETDA)**. Results show our method remains competitive under the more general substitutable setting and outperforms baselines on multiple deltas.
> >
> > Overall, these changes are reflected in the revised paper to address the reviewer’s concerns and improve clarity.

---

### Official Review · Reviewer_31vc · 2025-10-31

**Soundness:** 3
**Presentation:** 1
**Contribution:** 2
**Rating:** 6
**Confidence:** 4

**Summary:**

This paper considers a matching problem with the following characteristics.

- It is a many-to-one matching, where each players (the left side) select arms (right side).
- Players $i$'s preference for arm $j$ is $\mu_{i,j}$, which is unobserved.
- Arm $j$'s preferences over arms are $Ch_j(S)$ over sets of players $S$, satisfying substitution, rather than the stricter responsiveness notions.

The algorithm they present has three phases. The first is the indexing phase, where every player obtains an index number as soon as they are uniquely accepted by an arm. The second performs some type of arm indexing. However, it is not clear from the text why those steps are performed. Are they necessary, or do they simply make the main algorithm (Alg. 3) simpler to analyse? It overall appears unnecessarily complex to me. Perhaps there is a better solution.

That said, given that this scenario is a slight generalisation, the regret bounds are welcome, if not particularly groundbreaking. The proof of Phase 3 seems correct, but I could not really follow Phase 1 or 2's reasoning.

**Strengths:**

- Slight generalisation of existing work
- Player-optimal rather than pessimal bounds given

**Weaknesses:**

- It is unclear why this algorithm structure is chosen. It is not motivated at all, and phases 1 and 2 are not well explained.
- The proposal setting is a bit unusual, in that it is neither centralised mathcing, nor independent, but players sequentially pull arms in some instances.

**Questions:**

Better explain phase 1 and 2.

---

> ### Author Response · Authors · 2025-11-23
>
> We greatly appreciate your careful reading and constructive comments. Below, we will further explain and clarify the two key questions you raised: the necessity of Phases 1 and 2, and the setting of the proposal mechanism.
>
> First, regarding the necessity of Phase 1. The fundamental purpose of this stage is to construct a coordinable public order in a market where information cannot be exchanged and only substitutable preferences exist. Because under substitutable preferences, whether an arm $a_j$ accepts player $p_i$ depends on the set $S$ of all proposers to that arm in that round. Therefore, when a player is rejected, the feedback is highly ambiguous—the player cannot determine whether it is because the arm would never accept choosing them in the first place, or because in this round, the arm would not accept the combination proposed by them and other players. If, under this setting, no index is assigned to the player as a reference, it is difficult for any effective coordination to be formed among the players, and the player will be trapped because they cannot know the reason for being rejected by the arm, thus making it difficult to support the exploration and convergence in the subsequent Phase 3. In addition, most existing method directly assume an index for each arm and designs allocation mechanisms based on this. However, in many practical scenarios, such assumptions are unrealistic. We often only know the arm's name but cannot directly assign an index known to all players to different arms. Therefore, we introduce Phase 1, allowing all players and all arms to obtain a unique index. This not only eliminates the dependence on pre-defined arm index but also provides a prerequisite for all subsequent decentralized coordination steps. It is a key technical prerequisite for subsequent algorithms, not just an additional structure added for ease of analysis.
>
> Secondly, regarding the motivation and role of Phase 2. After obtaining the player and arm index in Phase 1, Phase 2 mainly utilizes the contrapositive of substitutability preference: if a player $p_i$ is rejected when proposing to arm $a_j$ alone, then according to the contrapositive of substitutability, $p_i$ will inevitably be rejected regardless of which other players propose to that arm in the future. This means that for $p_i$, this arm is destined to fail in any case. Based on this, in Phase 2, each player is arranged to explore all arms individually in a finite number of rounds using the index obtained in Phase 1. By having each player explore each arm individually, the set of arms that the player can explore is determined, avoiding the waste of a lot of time in Phase 3 on exploring arms that are bound to fail. In the latter half of Phase 2, instead of allocating a separate round for each explorable player-arm relationship, we prove that there exists a conflict-free exploration sequence of at most $2\cdot\max(\hat N',\hat K')$, such that each player can try each of its explorable arms at least once in this sequence. Thus, in the third stage, during the explore-then-commit process, as long as players select arms sequentially according to this sequence, they can obtain effective, collision-free feedback, significantly reducing the number of wasted runs. This achieves the optimal stable regret upper bound of $\mathcal{O}(\max(N',K')\log T/\Delta^2)$ given in Theorem 5.3. In other words, the second stage is not merely for simplifying the analysis, but directly determines whether the third stage can achieve a systematic and conflict-free exploration of all explorable arms within a controllable time window.
>
> Finally, regarding the question about proposal setting, our description here was indeed not clear enough. Our mechanism operates synchronously in rounds: in each round, all players independently select an arm and submit a proposal based on their own index and pre-published rules. The arm then determines its acceptance set for that round based on its own $Ch_j(\cdot)$, and all players jointly observe the matching results of that round. The role of the index is solely to construct an executable exploration sequence: in the absence of a centralized scheduler, each player only needs to know their own index and the publicly available construction rules to locally calculate which arm to try in round $t$. Therefore, this mechanism differs from completely independent and uncoordinated exploration, as well as centralized matching where actions are directly assigned by a central platform. More accurately, it is an implicit decentralized coordination based on the index and historical matching results. Our analysis shows that this minimal coordination introduces only a controllable $\max(N',K')$ factor, does not change the logarithmic magnitude of regret with respect to $T$, and maintains optimality of the same order of magnitude as existing work.

---

> ### Author Response · Authors · 2025-12-02
>
> Since no further follow-up was received from reviewers during the discussion phase, to facilitate the AC's independent completion of the final evaluation, we have summarized the key clarifications regarding the necessity of the phase design and proposal settings as follows:
>
> **1. Clarification of the necessity of Phase 1:**  Under **no communication and substitutable preferences**, acceptance depends on the round’s proposal set, making rejection feedback ambiguous. Phase 1 learns a consistent player/arm index via random proposals, resolving this ambiguity and enabling coordinated, conflict-free exploration.
>
> **2. Explanation of the necessity of Phase 2:** Using the index of Phase 1, Phase 2 applies the contrapositive of substitutability to filter infeasible arms and **builds a deterministic conflict-free exploration schedule**, ensuring each player tests every explorable arm **at least once** and enabling Phase 3 to achieve the player-optimal stable regret bound.
>
> **3. Clarifying the information structure and decentralization in our proposal setting:** In our setting, players submit proposals independently, each arm accepts according to its own choice function, and the matching outcome is publicly observable. Coordination is induced implicitly by **the shared public history and rules**, rather than by any centralized assignment.
>
> Overall, our three-stage structure is designed for a more general and challenging information structure of **substitutable preferences + no pre-defined index**; the revised draft has more explicitly incorporated the above motivations and mechanisms into the main text to improve readability and facilitate AC's evaluation of contribution and correctness.

---

### Meta-Review · Area_Chair_UA3Z · 2026-01-06

**Summary:**

This paper studies bandit learning in many-to-one matching markets where players have unknown (stochastic) preferences over arms, while arms have set-based preferences over groups of players satisfying substitutability (so stable matchings exist).
The goal is to minimize player-optimal stable regret, i.e., to learn enough about player rewards so that play converges to the player-optimal stable matching and regret is measured relative to that outcome.
To do so, the paper proposes a new method called, RIFLE, which has a three-phase procedure:
(i) a randomized “indexing” phase to assign a public ordering to players and arms without assuming a pre-defined arm index,
(ii) a phase that identifies which arms are individually explorable for each player (accepted when the player applies alone) and constructs a conflict-free exploration schedule, and
(iii) an explore-then-commit phase that estimates player preferences over the explorable set and then runs a Gale–Shapley-style proposal process to reach a stable outcome.
The paper provides the regret bound for the player-optimal stable matching under substitutable preferences and experimental comparisons.

**Reviewer Concerns:**

Among the reviewers, there were concerns regarding the following aspects:
(i) motivation and clarity of the phase design remain a central weakness,
(ii) "decentralized" claim depends on strong information assumptions.
(iii) possible "incremental" novelty relative to prior matching-bandit pipelines,
(iv) experimental evaluation is still relatively narrow.

**Reviewer Scores:**

Given the written discussion (with no reviewer follow-up after the rebuttal), I would not expect substantial score movement:

Reviewer 31vc (score 6): The rebuttal directly addresses the “why Phases 1–2?” question and clarifies the synchronous proposal mechanism. This might increase their understanding and possibly presentation assessment, but their main concern of “unnecessarily complex,” “not particularly groundbreaking” results likely remains. Expected score change: no change.

Reviewer QDsS (score 4): The added baseline(s) partially address the experimental criticism, but their concerns about (a) limited novelty and (b) reliance on globally observed matching outcomes are not fully resolved by rebuttal alone. Expected score change: no change

Reviewer aqJY (score 4): The rebuttal provides a substitutability-vs-responsiveness example and clarifies positioning versus ODA/AETDA, plus suggests added experiments. However, their core concern of “technical contribution rather limited / similar pipeline” likely persists. Expected score change: no change

---

### Decision · Program_Chairs · 2026-01-26

Reject